# Structural journey of an insecticidal protein against western corn rootworm

Guendalina Marini[1,2], Brad Poland[3], Chris Leininger [3,5], Natalya Lukoyanova[1], Dan Spielbauer[3], Jennifer K. Barry[3], Dan Altier[3], Amy Lum[3,6], Eric Scolaro[3], Claudia Pérez Ortega[3,7], Nasser Yalpani[3,8], Gary Sandahl[3], Tim Mabry[4], Jeffrey Klever[3], Timothy Nowatzki[3], Jian-Zhou Zhao [3], Amit Sethi [3], Adane Kassa [3], Virginia Crane[3], Albert L. Lu[3], Mark E. Nelson [3] ✉, Narayanan Eswar [3] ✉, Maya Topf [1,2] ✉ & Helen R. Saibil [1] ✉

The broad adoption of transgenic crops has revolutionized agriculture. However, resistance to insecticidal proteins by agricultural pests poses a continuous challenge to maintaining crop productivity and new proteins are urgently needed to replace those utilized for existing transgenic traits. We identified an insecticidal membrane attack complex/perforin (MACPF) protein, Mpf2Ba1, with strong activity against the devastating coleopteran pest western corn rootworm (WCR) and a novel site of action. Using an integrative structural biology approach, we determined monomeric, pre-pore and pore structures, revealing changes between structural states at high resolution. We discovered an assembly inhibition mechanism, a molecular switch that activates pre-pore oligomerization upon gut fluid incubation and solved the highest resolution MACPF pore structure to-date. Our findings demonstrate not only the utility of Mpf2Ba1 in the development of biotechnology solutions for protecting maize from WCR to promote food security, but also uncover previously unknown mechanistic principles of bacterial MACPF assembly.

Pore-forming proteins (PFPs) are a major class of soluble proteins that oligomerize into rings and undergo large irreversible conformational changes to become transmembrane assemblies and breach the membrane of target cells[1]. The unregulated ionic influx/efflux through the resulting pore cause serious osmotic imbalance and toxicity that eventually lead to cell death by necrosis[2,3].

PFPs derived from the entomopathogenic bacterium *Bacillus thuringiensis* (Bt) have been the basis for transgenic insect-resistant crops, benefiting growers globally for almost three decades[4–6]. *In planta* delivery of insect protection has proven to be highly effective and has environmental and economic benefits where deployed[7,8]. One

of the most devastating pests affecting maize production in North America and Europe and leading to significant yield loss is the coleopteran western corn rootworm (WCR; *Diabrotica virgifera virgifera*)[9,10]. Since the early 2000s, this pest has been effectively managed in North America by transgenic maize crops expressing Bt proteins from the Cry3 class (mCry3Aa, Cry3Bb1, and eCry3.1Ab) or Gpp34Ab1/Tpp35Ab1 (previously known as Cry34Ab1/Cry35Ab1)[11,12]. However, these proteins have encountered instances of reduced efficacy due to the development of resistance in field populations of WCR[8]. Since the discovery rate of useful Bt-derived insecticidal proteins seems to be slowing[13,14], it is imperative to search for novel

[1]Institute of Structural and Molecular Biology, Birkbeck, University of London, Malet St, London WC1E 7HX, UK. [2]Centre for Structural Systems Biology (CSSB), Leibniz-Institut für Virologie (LIV), Universitätsklinikum Hamburg-Eppendorf (UKE), Hamburg, Germany. [3]Corteva Agriscience, Johnston, IA 50131, USA. [4]Corteva Agriscience, Ivesdale, IL 61851, USA. [5]Present address: Syngenta, Research Triangle Park, NC 27709, USA. [6]Present address: Willow Biosciences, 319 N Bernardo Ave #4, Mountain View, CA 94043, USA. [7]Present address: Hologic, Inc., 250 Campus Drive, Marlborough, MA 01752, USA. [8]Present address: Dept. of Biology, University of British Columbia Okanagan, 3187 University Way, Kelowna, BC V1V 1V7, Canada. ✉e-mail: mark.e.nelson@corteva.com; narayanan.eswar@corteva.com; maya.topf@cssb-hamburg.de; h.saibil@bbk.ac.uk

proteins from non-Bt organisms with high activity and new modes/sites of action.

Here, we identified an insecticidal protein, Mpf2Ba1, active against WCR, and we carried out a comprehensive structure-function characterization of a PFP from a non-Bt organism, revealing key steps of pore formation. We demonstrate that gut fluid extracted from WCR larvae provides the necessary and sufficient environment for protein activation, triggering conversion from the monomer to the pre-pore state and arming it for membrane penetration as it converts into the pore state. Using X-ray crystallography and cryo-EM, we reveal all three structural states.

## Results

Mpf2Ba1 is a 53.2 kDa protein isolated from *Pseudomonas monteilii* through a multi-step purification process informed by artificial diet bioactivity screening against WCR larvae. In artificial diet bioassays, purified recombinant Mpf2Ba1 and a variant bearing nine conservative mutations (var 1167; Supplementary Table 1) showed strong activity against WCR and northern corn rootworm (NCR; *Diabrotica barberi*) larvae, but no activity against other major agricultural pest species (corn earworm, *Helicoverpa zea*; fall armyworm, *Spodoptera frugiperda*; European corn borer, *Ostrinia nubilalis*; soybean looper, *Chrysodeixis includens*; and southern green stinkbug, *Nezara viridula*) under the conditions tested (≥400 ppm), suggesting that the protein has a certain degree of specificity to coleopteran insect pests (Fig. 1a and Supplementary Table 2). In addition, Mpf2Ba1 exhibited specific binding to WCR midgut tissue, caused a rapid increase in ion permeability in a southern corn rootworm cell line, suggesting pore formation, and provided strong root protection from corn rootworm damage when expressed in transgenic maize tested under greenhouse

and field conditions (Fig. 1b–d, Supplementary Fig. 1, and Supplementary Text). Finally, artificial diet bioassays also show that Mpf2Ba1 is active against a field-derived population of WCR that exhibits signs of resistance to currently commercialized insecticidal proteins (mCry3A, and Gpp34Ab1/Tpp35Ab1; Supplementary Table 3 and Supplementary Results). This indicates that Mpf2Ba1 elicits WCR mortality through a novel site of action and can bypass resistance against existing commercial trait proteins.

### Structure of Mpf2Ba1-1167 soluble monomer

We first solved the X-ray structure of Mpf2Ba1-1167 at 2.1 Å resolution (Fig. 2a, Supplementary Fig. 7a, and Supplementary Table 4). The protein is comprised of two domains—an N-terminal MACPF domain and a C-terminal β-prism domain. The MACPF domain has a central four-stranded antiparallel β-sheet with a characteristic ~90° bend. The β-sheet is interrupted by three insertions: two sets of transmembrane hairpins emerge from its base (TMH1, buried inside the hydrophobic core, and TMH2, facing the surface), and a helix-turn-helix (HTH) motif, harboring several residues that are highly conserved in all known MACPF/CDCs[15] (Supplementary Figs. 2, 3), intercalates at the bend of strand β4. Both TMHs comprise a cluster of two α-helices and a loop, similar in length to the short TMHs of mammalian perforin-2 (PFN2, PDB IDs: 6U23 and 6SB3)[16,17].

The C-terminal domain adopts a β-prism fold, first identified as one of the three domains of insecticidal δ-endotoxins derived from Bt[18,19]. It is structurally closely related to the C-terminal domain of Mpf1Aa1 from *Photorhabdus luminescens* (PDB ID: 2QP2, 26% seq. id; 1.58 Å RMSD over 114aa out of 147aa)[20] and, to a lesser extent, to the membrane-associated β-prism domain of the multivesicular body subunit 12B (MABP-MVB12B) of human ESCRT-I complex

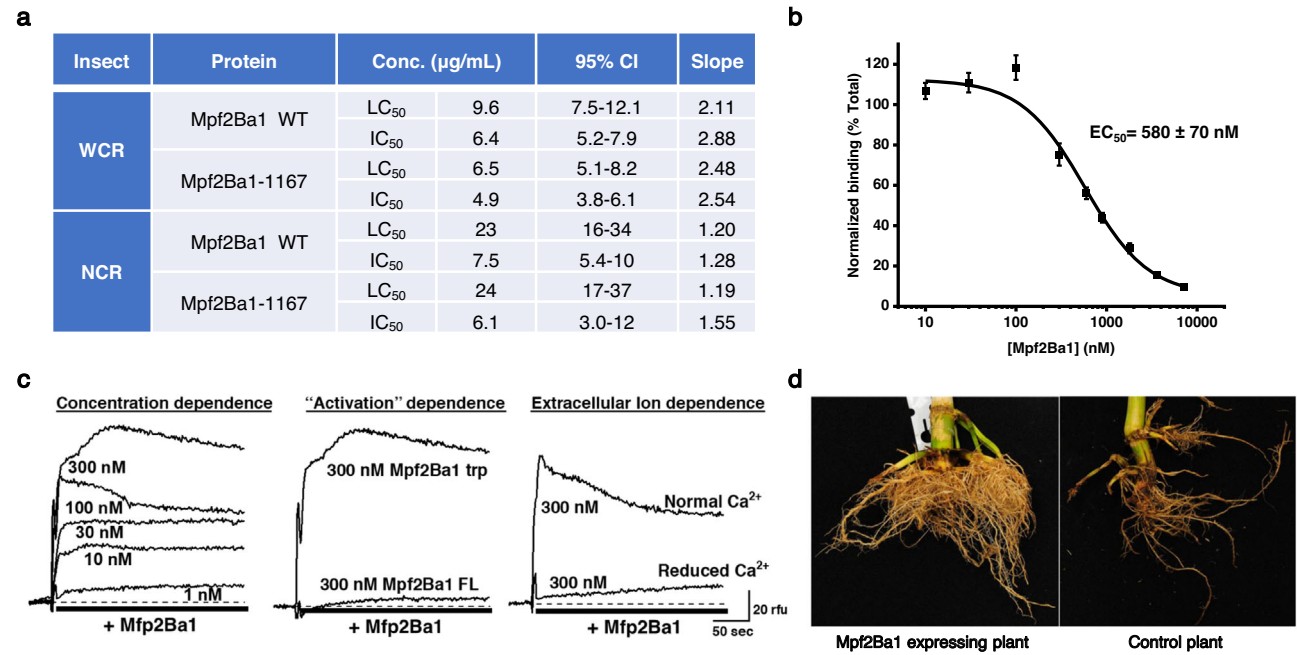

**Fig. 1 | Activity of Mpf2Ba1 against corn rootworm. a** Bioactivity of Mpf2Ba1 WT and Mpf2Ba1-1167 against WCR and NCR larvae in artificial diet bioassay. $LC_{50}$ and $IC_{50}$ (Methods) are similar for both proteins, with WCR exhibiting greater lethal sensitivity (lower $LC_{50}$ value) than NCR. Confidence intervals (95% CI) and the slope of the fitted curve from the analysis are also provided. **b** Specific binding of Mpf2Ba1 to WCR BBMVs is demonstrated by homologous competition of Alexa-Mpf2Ba1 by its unlabeled form. The values reflect the mean ± SEM from four observations of densitometry values determined from in-gel fluorescence images. **c** Real-time fluorescence responses of southern corn rootworm cells (Du182A) loaded with the $Ca^{2+}$-sensitive Fluo-3 dye are shown. The cells were exposed to

Mpf2Ba1 WT at different concentrations after trypsin activation in normal media (left), to 300 nM Mpf2Ba1 before (FL) and after trypsin activation (trp) in normal media (middle), and to 300 nM Mpf2Ba1 after trypsin activation in normal $Ca^{2+}$ or nominally $Ca^{2+}$-free media to demonstrate that the signal required influx of extracellular $Ca^{2+}$ (right). **d** Pictures of corn roots showing feeding damage by WCR from a field study reported in Supplementary Fig. 1b that included plants expressing Mpf2Ba1 and plants with the same genetic background that did not express Mpf2Ba1 (control plant). These plants were in the Johnston, IA location. Source data for panels **a**–**c** and testing against Lepidoptera and *Nezara viridula* are provided within the Source Data file.

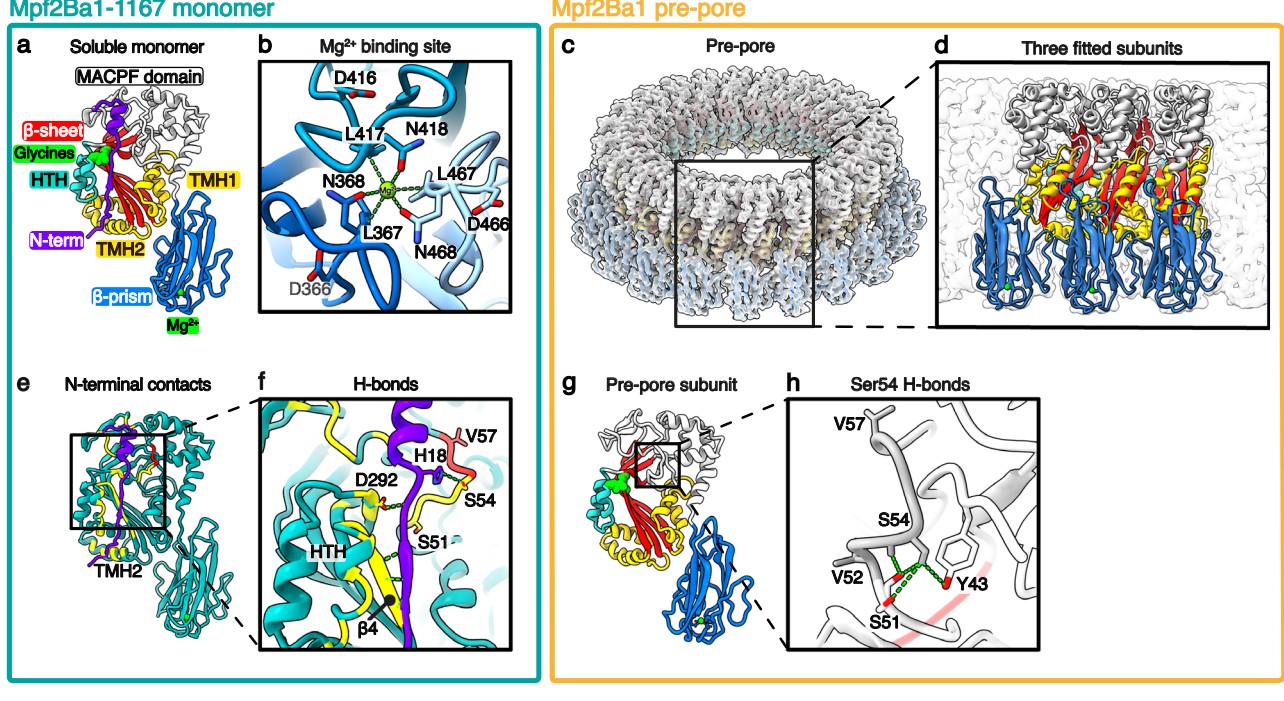

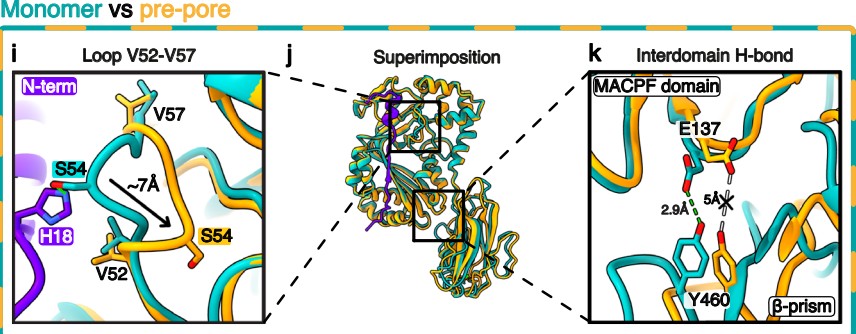

**Fig. 2 | Structures of the Mpf2Ba1-I167 monomer, Mpf2Ba1 wild-type pre-pore, and their comparison. a** X-ray structure of Mpf2Ba1-I167 soluble monomer. **b** View of the C-terminal membrane binding site showing conserved Asp, Leu, and Asn (DLN) residues in the β-prism, and residues Leu and Asn coordinating a Mg²⁺ cation in the center. **c** Single particle cryo-EM map of the 22-mer pre-pore at 3.1 Å resolution, with 22 fitted pre-pore subunits. **d** Inset of three pre-pore subunits fitted in the pre-pore map. **e** Interactions between the 30 N-terminal residues (in purple) and surrounding residues (in yellow) are highlighted. **f** Inset of the N-terminal contacts with surrounding residues (green dashes). **g** Model of the wild-type pre-pore subunit depicted according to the domains. **h** Inset showing the buried position of loop Val52-Val57 in the pre-pore, where Ser54 contacts Tyr43 and Ser51, all conserved in bacterial MACPFs. **i**, Inset showing the comparison of loop Val52-Val57 positions between the soluble monomer (in cyan), and the pre-pore structure (in orange), where it is moved -7 Å towards the MACPF domain core. **j** Superimposition of Mpf2Ba1-I167 soluble monomer (in cyan) and wild-type pre-pore models (in orange). **k** Inset showing the -15° rotation of the β-prism breaking the H-bond between the conserved Tyr460-Glu137.

(PDB ID: 3TOW, 21% seq. id; 1.6 Å RMSD over 84aa out of 147aa) (Supplementary Fig. 3)[21]. In Mpf2Ba1, each of the three β-sheets of the β-prism, arranged around a local three-fold axis, contains a unique repeated motif: Asp-Leu-Asn (DLN). These DLN motifs create an octahedral geometry coordinating a Mg²⁺ at the symmetry axis through the Leu carbonyl oxygen and the Asn side-chain amide oxygen (Fig. 2b and Supplementary Fig. 2). Similar domains of other MACPF proteins, such as Mpf1Aa1[20], Mpf3Aa1, an insecticidal protein from *Chromobacterium piscinae* (PDB ID: 6FBM)[22], and PFN2[16,17], have Ca²⁺ as the coordinating ion. Structurally, the metal chelation seems to tuck the three β-prism loops into gaps between the adjacent β-sheets, increasing domain stability.

## WCR gut fluid-induced activation/oligomerization

A major finding that facilitated Mpf2Ba1 structural characterization is that gut fluid extracted from WCR larvae is necessary and sufficient to induce oligomerization in vitro in the absence of brush-border membrane vesicles (BBMVs) prepared from isolated gut tissue (Supplementary Fig. 4). Upon incubation in gut fluid, an N-terminal cleavage after Lys25, as determined through Edman degradation sequencing, and subsequent oligomerization step are triggered by interaction of monomers with yet unidentified midgut component(s). Based on the observations that heat-inactivated gut fluid or the organic-extractable phase of gut fluid do not induce oligomerization (Supplementary Fig. 5), we suspect that the oligomerization factor is a protein. Furthermore, we established that although N-terminal cleavage can be achieved using trypsin, WCR gut fluid is still necessary to trigger oligomerization. After gut fluid-induced oligomerization, Mpf2Ba1 monomers formed higher-order complexes that were isolated via size exclusion chromatography and imaged by cryo-EM. The 3D reconstruction of ring-shaped oligomers at 3.1 Å resolution revealed an intermediate state of Mpf2Ba1: a 22-mer pre-pore that is 80 Å in height, 260 Å in diameter, and encircles a 120 Å-wide lumen (Fig. 2c, d, Supplementary Fig. 6, and Supplementary Table 5).

## Pre-pore N-terminal cleavage

By fitting and refining the X-ray structure of Mpf2Ba1 into the pre-pore map, we discovered that we could model the entire protein in the density without breaks, except for the 30 N-terminal residues, five residues more than the 25 identified biochemically (Supplementary Fig. 7). The missing density for those residues could be due to flexibility or disorder of the N-terminal region, or to additional proteolytic cleavage. Our finding that N-terminal cleavage is also required for Mpf2Ba1 pore formation on insect cells in vitro (Fig. 1c) prompted us to search for proteases active on the putative cleavage site before Thr31 observed in the pre-pore structure. Aspartic proteases belonging to the family of cathepsins, that are abundant in midguts of WCR larvae[23], emerged as the highest-ranked candidates. Although further investigation is needed, a cathepsin-D-like protease, along with trypsin that would cleave at Lys25, could play a role in triggering Mpf2Ba1 activation via N-terminal cleavage in vivo.

## Mpf2Ba1 transition from monomer to pre-pore

Comparison of the pre-pore structure with the soluble monomer shows small but significant conformational differences. In the monomer, the N-terminus lies adjacent to one side of the MACPF domain, forming a short β-strand–β-sheet interaction with the central β-sheet, and two more H-bonds: between Asp292, stabilized by the conserved Pro291, and the N-terminal backbone, and between one of the nitrogen atoms of N-terminal His18 imidazole ring and the oxygen atom of Ser54, highly conserved among bacterial MACPFs (Fig. 2e, f and Supplementary Fig. 2). Upon N-terminal removal via gut fluid activation, these contacts are lost and residues on one side of the MACPF domain become fully exposed. Since we observed that some of these residues are involved in intermolecular contacts with the neighboring subunit, we think that the steric hindrance imposed by the N-terminus could prevent premature interactions/oligomerization of soluble monomers.

The MACPF domain in the Mpf2Ba1 pre-pore remains mostly unchanged, except for loop Val52-Val57 that moves from an external position, in contact with the uncleaved N-terminus, to a buried position in the oligomeric structure, as measured from the accessible surface area per residue (Fig. 2g–i and Supplementary Fig. 8). The X-ray density for this loop is clear, albeit weaker and with higher B-factors than the surrounding residues, indicating greater flexibility. Because residues of this loop are in contact with the neighboring subunit, it is likely that, upon N-terminal cleavage, this region rearranges and gets stabilized by the oligomeric conformation. Moreover, it seems that the loop is re-oriented by ~7 Å to bring the highly conserved Ser54 from a solvent-exposed position in the monomer, in contact with His18 of the N-terminus, to a core-buried position towards the MACPF domain in the oligomer. There it engages in H-bonds with Tyr43 and Ser51 (Fig. 2h), also highly conserved in bacteria (Supplementary Fig. 2). The Ser54 switch, triggered by N-terminal cleavage and/or oligomerization, may represent a key step in Mpf2Ba1 activation, as well as a general activation mechanism of bacterial MACPFs, due to the high conservation of the residues involved.

The pre-pore β-prism domain is rotated ~15° clockwise relative to its position in the monomeric structure. This slight rotation disrupts the only interdomain H-bond between MACPF and β-prism, involving the carboxyl oxygen of Glu137 and the hydroxyl group of Tyr460 (Fig. 2j, k, Supplementary Fig. 8, and Supplementary Movie 1). Based on residue conservation, this H-bond seems specific to bacterial MACPFs (Supplementary Fig. 2), and its loss could be part of an allosteric signal relayed from the C-terminal domain upon membrane binding, possibly associated with oligomerization. Interestingly, although AlphaFold2[24] accurately predicts Mpf2Ba1 domains and secondary structures of the soluble monomer and pre-pore, including the N-terminal α-helix, it does not fully predict either of these states, as the β-prism is not oriented precisely with respect to the MACPF domain in either state, nor does it predict the pore state (Supplementary Fig. 7).

## Pore conversion

The transition from pre-pore-to-pore requires an energetically costly structural rearrangement[25]. We were able to convert Mpf2Ba1 pre-pores into pores by either introducing lipids, in the form of POPC and 4% cholesterol mixed liposomes, or by raising the incubation temperature between ~50–53 °C without adding lipids. When using liposomes, we observed membrane-inserted structures that confirm the pore-forming activity of Mpf2Ba1 (Fig. 3a and Supplementary Fig. 4). Although heating is not the physiological trigger, it appears to catalyze the refolding of the α-helical TMHs into β-hairpins to form a β-barrel, which has higher thermal stability than the helical conformation[26], confirming that pre-pore-to-pore conversion is endothermic[27]. Heat-converted pores were assessed by negative-stain EM and optimized for cryo-EM single-particle analysis. The 2D classes of the pore showed a distribution of stoichiometries, ranging from 21- to 24-fold, demonstrating heterogeneity of oligomerization. It was also evident that Mpf2Ba1 pores dimerized tail-to-tail: the barrel of one pore was inserted inside the other, and the inner and outer pore had different stoichiometries (Supplementary Fig. 9, 3D classes in red and blue). Despite the most common stoichiometry in the 2D classes being a 22-mer, the inner ring of each dimer showed a consistent 21-fold symmetry and a clear structure, while the outer ring was a mixture of 22-fold and higher symmetries. Therefore, we decided to refine the 3D structure of the inner 21-mer pore, which was determined at 2.6 Å resolution and revealed a pore with an outer diameter of 240 Å, β-barrel diameter of 135 Å, and a height of 120 Å, increased by 40 Å from the pre-pore conformation (Fig. 3b, Supplementary Fig. 9, and Supplementary Table 5).

## Mpf2Ba1 transition from pre-poretopore

The map resolution was appropriate to build an accurate atomic model by rigid body fitting of the Mpf2Ba1 pre-pore model without TMHs and subsequent manual building as β-hairpins (Fig. 3c, d and Supplementary Fig. 7e–g), followed by full model refinement and validation (Supplementary Tables 4, 6).

In the pore conformation, the β-sheet at the MACPF core straightens by an average dihedral angle of 36° to allow the helical bundles to refold into membrane-spanning β-hairpins and collectively form a β-barrel (Fig. 3e, f and Supplementary Movie 2). The flexibility for the central β-sheet straightening is provided by glycines that are >95% conserved among MACPF/CDCs and function as a hinge (Supplementary Figs. 2, 3)[20,28].

The straightening of the MACPF central β-sheet repositions the HTH, which is pushed ~8 Å towards the lumen of the pore, restricting the diameter locally from 135 to 110 Å (Fig. 3f). This shift brings the HTH in close lateral contact with residues of the neighboring subunit, where Phe273 contacts Tyr92 on β1-strand in a π-π interaction, and the side-chain nitrogen of His270 forms a H-bond with the oxygen of Gln145 on β2-strand (Fig. 3g, h). A single H-bond was detected in the pre-pore, connecting neighboring HTHs (Fig. 3i, j). In the MACPF domain of pleurotolysin (PlyB), the HTH displacement was proposed as one of the key triggers for the pore conformational change[28]. In Mpf2Ba1, we see that HTH motion follows the straightening of the TMHs and does not move relative to them (Supplementary Movie 3), but the HTH intermolecular contacts change throughout pore formation. Based on the different interactions, we propose key regulatory roles for HTH in (i) preventing premature interaction between monomers in solution, (ii) controlling the transition from pre-pore-to-pore, and (iii) stabilizing inter-subunit interactions in the final pore structure[28–30].

The biggest conformational change during pore formation involves the two TMHs that unfurl from α-helical bundles into ~100 Å-long membrane-spanning β-hairpins, a unique feature of the MACPF/CDC superfamily (Fig. 3b–e and Supplementary Movies 2, 3)[31]. As in other MACPF structures, each subunit of Mpf2Ba1 pore contributes four β-strands to the β-barrel (84 β-strands for the Mpf2Ba1 21-mer).

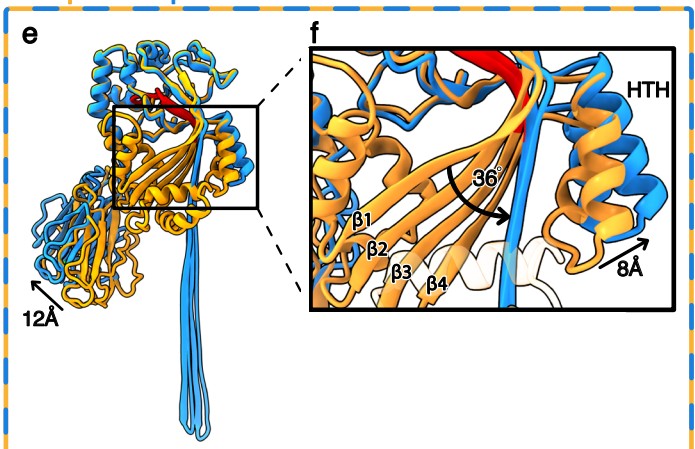

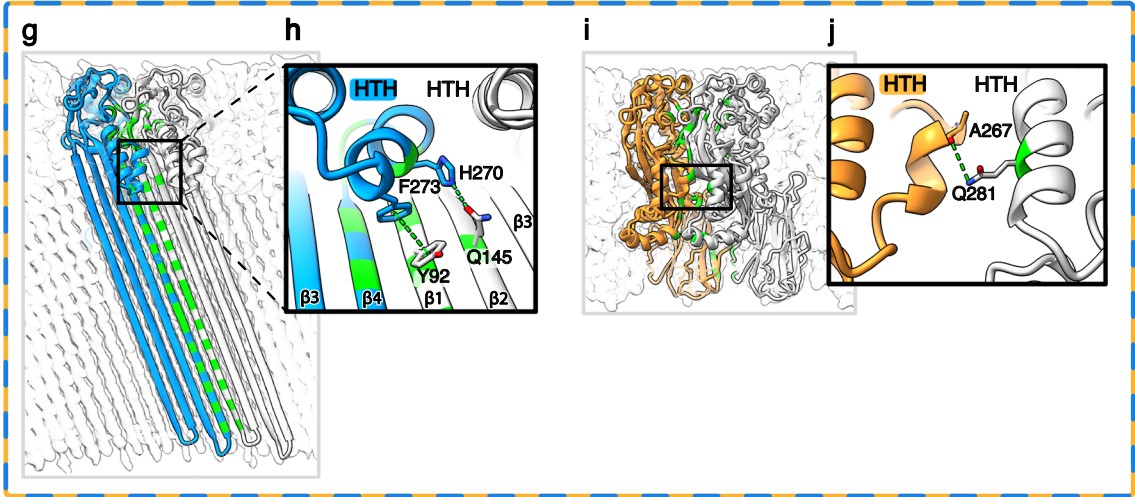

The β-strands are tilted by 20° from the membrane normal, in agreement with the β-barrel shear number $S = n/2$, where $n$ is the number of β-strands, found in giant β-barrel architecture[32,33]. The β-hairpins are stabilized by intra- and intermolecular H-bonds between β-strands, and the β-barrel is stabilized inside the membrane by aromatic residues frequently found in transmembrane β-barrels, oriented towards the polar phospholipid moiety and commonly reported as aromatic girdles[34–36]. In Mpf2Ba1, the polar residues Phe120 on strand β2 at the intracellular side of the membrane bilayer and Tyr219 on the β3-strand at the extracellular side are probably responsible for positioning and stabilizing the β-barrel inside the membrane. Lastly, the C-terminal β-prism domain moves ~12 Å outwards relative to its pre-pore position (Fig. 3e and Supplementary Fig. 8), possibly to allow the MACPF domain to approach and fully span the membrane.

**Fig. 3 | Mpf2Ba1 pore and comparison with the pre-pore structure. a** Liposome-inserted pores (white stars) imaged in negative-stain EM on the left and cryo-EM on the right. Scale bars: 50 nm; $n \geq 3$. **b** Single particle cryo-EM map of the 21-mer pore at 2.6 Å resolution and fitted pore subunits. **c** Inset of three pore subunits showing the aromatic residues Phe120 and Tyr219 (in green) forming an aromatic girdle around the β-barrel. **d** Model of the pore subunit. **e** Superposition of the pre-pore model (in gold) onto the pore model (in blue), revealing the conformational change between the two structures. The MACPF core signature motif is highlighted in red.

**f** Inset showing the straightening of the MACPF central sheet, where the strands β1–4 align towards the center of the pore by an average dihedral angle of -36°, and the HTH moves by 8 Å into the pore lumen. **g** Two neighboring pore subunits showing all intermolecular hydrogen bonds (in green). **h** View of the intermolecular contacts of HTH and strands β1 and β2 of the neighboring subunit. **i** Two neighboring pre-pore subunits showing all intermolecular H-bonds (in green). **j** View of the single intermolecular hydrogen bond between neighboring HTHs.

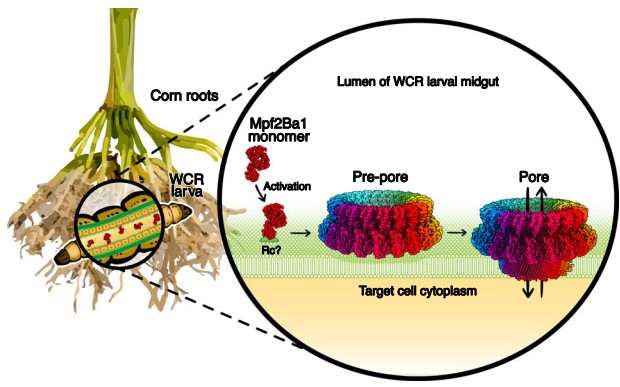

**Fig. 4 | Proposed model of Mpf2Ba1 insecticidal activity against WCR larvae.** The model shows our proposed in-plant protection mechanism in the context of corn roots. WCR larvae feeding on Mpf2Ba1-protected corn ingest Mpf2Ba1 monomers expressed by plant root tissues. The soluble monomers reach the larval midgut and are activated by a gut protease before/while engaging a not-yet-identified receptor (Rc) on the target cell and oligomerize into pre-pores. Pre-pores convert into irreversible transmembrane pores that allow uncontrolled ion flux across the plasma membrane, disrupting cellular function and causing the larvae to die of starvation.

In addition to inter-subunit interactions, the nonuniform surface charge distribution of PFP oligomers seems to contribute to subunit stabilization and membrane insertion. Indeed, neighboring Mpf2Ba1 pre-pore and pore subunits showed complementary charge interaction, as inferred from electrostatic potential maps. On one side of the MACPF domain, a negatively charged bulge and a positively charged pocket match exactly with a positively charged pocket and a negatively charged bulge on the neighboring subunit in both pre-pore and pore. Moreover, the charge distribution on the inner and outer surfaces of the pore is as expected for a transmembrane protein, with a highly polar lumen-facing surface, mostly water exposed, and a hydrophobic membrane-facing surface of the β-barrel forming an aliphatic nonpolar belt of approximately ~40 Å in height (Supplementary Fig. 10).

## Discussion

We have identified and characterized the non-Bt insecticidal protein Mpf2Ba1, providing mechanistic details for how it effectively kills the devastating maize pest WCR by forming pores in midgut membranes targeting a different binding site than the insecticidal proteins currently deployed in commercial transgenic maize hybrids (Fig. 4). We solved the Mpf2Ba1 X-ray structure and showed how N-terminal residues inhibit the association of monomers in the soluble form. We then demonstrated that proteolysis in the presence of gut fluid activates monomers to self-assemble into pre-pores, which we reconstructed by cryo-EM at 3.1 Å resolution. The subtle but significant changes between monomers and pre-pore subunits revealed molecular details related to target recognition and activation/oligomerization. The heat-induced conversion of pre-pores into pores allowed us to solve the 21-mer pore cryo-EM structure at 2.6 Å resolution, showing the dramatic conformational change of the pore subunits and their HTH-mediated inter-subunit stabilization.

Our work provides not only a high-resolution oligomeric structure for a MACPF insecticidal protein targeting WCR, but also the highest-resolution MACPF pore structure to-date. We demonstrate that Mpf2Ba1 efficacy and unique site of action give it great potential for protecting maize from WCR damage, particularly where there is resistance to current trait proteins.

## Methods

### Mpf2Ba1 and Mpf2Ba1-1167 LC$_{50}$/IC$_{50}$ determinations

Mpf2Ba1 or Mpf2Ba1-1167 protein was incorporated into an artificial diet to conduct bioassays on key rootworm species (WCR, NCR, and *Diabrotica speciosa*). Corn rootworm diet was prepared as described in ref. 37. The test involved six different protein doses plus buffer control with 32 observations for each dose in each bioassay. Neonate larvae were infested into 96-well plates containing a mixture of the Mpf2Ba1 proteins (5 μL/well) and diet (25 μL/well), each well with approximately five to eight larvae (<24 h post-hatch). After one day, a single larva was transferred into each well of a second 96-well plate containing a mixture of the Mpf2Ba1 (20 μL/well) and diet (100 μL/well) at the same concentration as the treatment to which the insect was exposed on the first day. The plates were incubated at 27 °C, 65% RH in the dark for 6 days. The 50% lethal concentration (LC$_{50}$) for polypeptides in the bioassay was calculated using the PROC PROBIT procedure in SAS software (v9.4; SAS Institute, Cary, NC, U.S.A.). Larvae were scored on a scale of 0–3 (unaffected, stunted, and severely stunted, respectively) or dead. Dead and severely stunted counts were pooled as total response for the calculation of the concentration for inhibition of 50% of the individuals (IC$_{50}$) using the same method (Fig. 1a).

### Mpf2Ba1 mode of action

To understand the mechanism of Mpf2Ba1 toxicity, the specific binding of the purified protein with WCR midgut tissue was evaluated by in vitro competition assays. Midguts were isolated from third-instar WCR larvae to prepare brush-border membrane vesicles (BBMVs) as described in ref. 37 using amino-peptidase activity to track enrichment. BBMVs represent the apical membrane component of the epithelial cell lining of insect midgut tissue and therefore serve as a model system for how insecticidal proteins interact within the gut following ingestion.

Purified Mpf2Ba1 was diluted to 1 mg/ml and processed with immobilized trypsin resin (Sigma, T1763 or Promega V901B) using resin at a 1:1 (v:v) ratio incubated at room temperature overnight. An aliquot of the trypsin-processed protein was then labeled with Alexa-Fluor® 488 (Life Technologies) and unincorporated fluorophore was separated from the labeled protein using buffer exchange resin (Life Technologies, A30006) following the manufacturer's recommendations. Prior to binding experiments, proteins were quantified by gel densitometry following Simply Blue® (Thermo Fischer Scientific) staining of SDS-PAGE resolved samples that included BSA as a standard.

To establish specific binding and to evaluate binding affinity, BBMVs (2.5 μg) were incubated with Alexa-labeled Mpf2Ba1 (Alexa-Mpf2Ba1; 5 nM) in binding buffer (50 mM sodium chloride, 2.7 mM potassium chloride, 8.1 mM disodium hydrogen phosphate, and 1.47 mM potassium dihydrogen phosphate, pH 7.5 containing 0.1%

Tween20®; 100 μL) for 1 h at RT in the absence and presence of increasing concentrations of unlabeled Mpf2Ba1 (0.01–7.2 μM). Centrifugation at 20,000×*g* was used to pellet the BBMVs to separate unbound Alexa-Mpf2Ba1 remaining in the solution. The BBMV pellet was then washed twice with binding buffer to eliminate the remaining unbound Alexa-Mpf2Ba1. The final BBMV pellet (with bound fluorescent protein) was solubilized in reducing Laemmli sample buffer, heated to 100 °C for 5 min, and subjected to SDS-PAGE using 4–12% Bis-Tris polyacrylamide gels (Life Technologies). The amount of Alexa-Mpf2Ba1 in the gel from each sample was measured by a digital fluorescence imaging system (ImageQuant™ LAS4000−GE Healthcare). Digitized images were analyzed by densitometry software (Phoretix™ 1D, TotalLab, Ltd.). Figure 1b shows that Mpf2Ba1 binds specifically to WCR BBMVs. The densitometry values were normalized to the signal observed for Alexa-Mpf2Ba1 binding in the absence of competitor ("Total") and plotted vs the concentration of unlabeled protein present during competitions. The resulting distribution was then fit to a logistic equation in OriginPro software (v. 2021b; Originlab Corp., Northampton, MA, USA); $y = A2 + (A1 - A2)/(1 + (x/x_0)^p)$, where A1 and A2 are the maxima and minima, respectively, x is the unlabeled protein concentration, and $x_0$ and $p$ corresponds to the $EC_{50}$ and slope values, respectively.

## Pore formation assay

To evaluate the ability of Mpf2Ba1 to undergo pore formation, a cell line derived from *Diabrotica undecimpunctata* (southern corn rootworm) designated IPLB-Du182A (Du182A) was used[38]. The Du182A cell line was obtained from the United States Department of Agriculture, Agriculture Research Service (Beltsville Agriculture Research Center, Beltsville, MD, USA). It is worth noting that southern corn rootworm larvae did not exhibit susceptibility to Mpf2Ba1 protein (1250 ppm) under the conditions tested in artificial diet bioassays (Source Data File). The reasons for this continue to be explored. Du182A cells were maintained in culture by growing in SF-900 II medium (Gibco, 10902) supplemented with 3% heat-inactivated FBS (Gibco, 16140) and antibiotics (Gibco, 5240) in T-75 flasks at 27 °C. To detect pore formation, Du182A cells were grown in 96-well assay plates (Corning, 3594). After reaching approximately 70% confluence, the medium was removed, and the cells were washed with a serum-free medium. The cells were then loaded with Fluo-4 AM (10 μM from DMSO stock; Thermo Fisher Scientific, F14201) in medium consisting of 50% Hank's buffered saline (HBSS; Gibco 14025092) supplemented with 3.7 mM $CaCl_2$ (final concentration of 5 mM) and 50% serum-free SF-900 II medium along with 2.5 mM probenecid (Sigma, P8761) and 0.2% (v:v) Pluronic F-127 (Thermo Fisher Scientific, P6867) for 1 h at 27 °C. The loading solution was then removed and replaced with an assay medium consisting of 50% HBSS supplemented with $CaCl_2$ (5 mM final concentration), 50% serum-free SF-900 II and 2.5 mM probenecid. The loaded cells were then placed in a Flexstation 3 (Molecular Devices) along with a compound plate that contained appropriate working stock dilutions of Mpf2Ba1 proteins. Mpf2Ba1 (1.6 mg/ml) was activated by treating the full-length protein with immobilized trypsin beads (Sigma T1763) 1:1 (v:v) at 25 °C until no full-length protein was detected (>30 h). Testing for pore formation was achieved by the transfer of activated Mpf2Ba1 protein (unless otherwise stated) from the compound plate to the assay plate using the Flexstation's built-in fluidics module and real-time monitoring of fluorescence intensity controlled by SoftmaxPro (v5.2, Molecular Devices) running in Flexmode. The transfer of Mpf2Ba1 was carried out after monitoring baseline fluorescence for 30 s (sampling interval ~1 s) and the cell responses were monitored for an additional 4.5 min. To construct figures, data were exported into OriginPro (v.2021b), where baseline fluorescence was subtracted from each well, and the subtracted responses were plotted versus elapsed time.

## Cross-resistance to commercial trait proteins

The activity of Mpf2Ba1 and Mpf2Ba1-1167 were evaluated in artificial diet bioassays using a WCR population collected from a research location in Readlyn, IA, which is situated within an area where there is high use of commercial hybrids expressing proteins that control damage from rootworms (Cry3 class and Gpp34Ab1/Tpp35Ab1). This population of WCR has been designated the "Readlyn" strain and has signs of resistance to both mCry3A and Gpp34Ab1/Tpp35Ab1 (see Supplementary Table 3). With this in mind, the insecticidal activities of the Mpf2Ba1 proteins against the Readlyn population were compared to their activity against a susceptible non-diapausing laboratory population of WCR. The format of the bioassays was similar to the method described above for bioactivity assessments against susceptible WCR larvae with certain differences. There were five to eight protein concentrations plus buffer control with 24 observations for each dose. Mortality data were analyzed following PROC PROBIT procedure in SAS software v9.4 for obtaining $LC_{50}$ and 95% confidence interval.

## Crystal growth and data collection

Crystals of Mpf2Ba1-1167 were grown by hanging drop vapor diffusion method at 22 °C. Crystals of Mpf2Ba1-1167 were obtained by a 1:1 ratio of 10 mg/ml protein solution and reservoir solution containing 0.2 M $MgCl_2$ hexahydrate, 0.1 M HEPES pH = 7.5 and 30% PEG 400. Crystals were flash frozen in liquid $N_2$ and mounted on a Rigaku Micromax-007 HF X-ray source at Iowa State University Macromolecular X-ray Crystallography facility. Data were collected using an R-Axis IV++ image plate detector. Mpf2Ba1-1167 crystals diffracted to 2.1 Å and belong to space group $P4_12_12$ with one molecule in the asymmetric unit. Diffraction data were indexed and integrated with iMOSFILM and scaled with SCALA (CCP4)[39].

## Structure determination of Mpf2Ba1-1167

The atomic structure of Mpf2Ba1-1167 was solved using the molecular replacement program PhaserMR (CCP4). The structure of the Mpf1Aa1protein from *Photorhabdus luminescens* (PDB ID: 2QP2)[20] was used as the search model. A suitable solution for the rotation and translation functions was identified. The sequence for the Mpf2Ba1-1167 was then built into the electron density using WinCoot©[40]. The model was refined using Refmac5[41] from CCP4[39] to an *R*-factor = 0.236 and *R*-free = 0.267 with >96% of amino acids in allowed regions of the Ramachandran Plot.

## Sequence analysis

The degree of Mpf2Ba1 sequence conservation in bacteria was obtained using the ConSurf server[42]. ConSurf defines a conservation score based on the evolutionary rate of a particular position through alignment and phylogenetic analysis of homologous sequences. A total of 249 sequences from a BLAST™[43] search in the nonredundant database were used to derive normalized conservation scores. The conservation scores were distributed into nine grades, from the most variable 1 to the most conserved 9. The generated alignment file was read into UCSF ChimeraX v1.4[44] and represented on the Mpf2Ba1 structure, with the corresponding residues colored according to the scores (Supplementary Fig. 2).

The sequence/structure conservation for Mpf2Ba1 was obtained running HHPred[45], a method for sequence database searching and structure prediction very sensitive in finding remote homologs, on the PDBmmCif30 database, based on the Protein Data Bank (PDB)[46] and with a maximum sequence identity of 30%, for the sequence of the MACPF domain (residues 31–331) and for the β-prism domain (residues 338–484) separately.

**Oligomer formation and purification for structure determinations.** Soluble monomeric Mpf2Ba1, purified at 6 mg/mL stock in 1x PBS

buffer (Gibco, 10010056), was diluted with an equal volume of freshly thawed WCR gut fluid (GF). GF was prepared from freshly dissected larval guts (third instar) mixed with 1x PBS (5 L/gut) incubated on ice for 90 min and then centrifuged at 20k×*g* for 15 at 4 °C to collect the supernatant. The Mpf2Ba1/GF mixture was incubated for 1–4 h at 37 °C with rotary mixing. To purify oligomers after WCR GF treatment, the incubated suspension was loaded onto a Superose 6 Increase 10/300 GL size exclusion column (GE Healthcare Biosciences, Piscataway, NJ, USA), pre-equilibrated with 1x PBS pH 7.4 (Gibco, 100100) filtered using a 0.22 μm filter and degassed. Protein was eluted with the same buffer at 0.5 mL/min and fraction size 0.5 mL for 1.5 column volumes. For vitrification and cryo-EM imaging, fractions containing Mpf2Ba1 pre-pores were pooled and concentrated to ~0.8 mg/mL using a centrifugal filter 100 kDa cutoff (Sigma-Aldrich).

Mpf2Ba1 pores were obtained by incubating purified pre-pores, or a 1:1 mixture of Mpf2Ba1 soluble monomers and GF, with liposomes (described below) for 1 h. Alternatively, pores were obtained by sonication in a water bath or by incubation at ~50–53 °C for 1 h without adding lipids, the latter yielding the best pre-pore-to-pore conversion. Subsequent size exclusion chromatography onto a Superose 6 Increase 10/300 GL size exclusion column (GE Healthcare Biosciences, Piscataway, NJ, USA), elution buffer as described above-separated pores from monomers and/or bigger liposomes, if present. Fractions containing Mpf2Ba1 pores were pooled and concentrated to ~0.6 mg/mL using a centrifugal filter 100 kDa cutoff (Sigma-Aldrich) and then vitrified and imaged by cryo-EM.

### Cryo-EM sample preparation
Freshly purified Mpf2Ba1 pre-pores concentrated to ~0.8 mg/mL and pores concentrated to ~0.6 mg/mL (3 μl) were applied to graphene oxide coated holey carbon grids (EMS, CFlats R 1.2/1.3 and 2/1 300 copper mesh) prepared following the procedure of ref. 47. After 5 s blotting time, samples were plunge-frozen in liquid ethane cooled by liquid nitrogen[48], using a Vitrobot Mark IV (Thermo Fisher Scientific) and stored under liquid nitrogen until use. Screening of cryo-EM conditions was performed on a 200 kV F20 (Thermo Fisher Scientific) with a DE-20 direct electron detector camera (DirectElectron).

### Data collection and image processing of pre-pores
A dataset of 3578 movies was collected on a 300 kV Titan Krios microscope (Thermo Fisher Scientific) with a K2 Summit direct electron detector camera and Quantum energy filter (Gatan) using EPU v4 (Thermo Fischer Scientific).

Image processing and single-particle analysis were performed in cryoSPARC v2.9.0 (Structura Biotechnology)[49], unless otherwise stated (Supplementary Fig. 6). Electron micrograph movie frames were aligned using cryoSPARC patch motion correction. CTF parameters were estimated, refined, and corrected using cryoSPARC patch CTF estimation. Particles were picked and subjected to several rounds of two-dimensional (2D) classification, selecting classes with identifiable secondary structure features. After manual curation to exclude low-quality micrographs based on poor CTF resolution (>9 Å) and particles with deviated local motion trajectories, a total of 64,428 particles were selected and used to generate an ab initio three-dimensional (3D) model without imposed symmetry (C1). The map revealed 22-mer rings forming tail-to-tail dimers. These double-ring structures have been observed in other pore-forming proteins, probably as a consequence of two membrane-binding surfaces interacting with each other[16]. This ab initio map was used as an initial model for homogeneous refinement. D22 symmetry was imposed in the refinement of the pre-pore, since particles, 2D class averages and ab initio map showed 22-mer pre-pores paired in dimers. The final refinement was carried out using nonuniform refinement with per-particle CTF refinement. The final map was sharpened using the DeepEMhancer v0.14 "tightTarget" option from the unfiltered unmasked halfmaps[50].

The resolution of the map was determined with masking-effect corrected Fourier Shell Correlation (FSC) as implemented in cryoSPARC. Local resolution was estimated in the final map using "blocres"[51], also implemented in cryoSPARC. The pre-pore image processing workflow and parameters are summarized in Supplementary Fig. 6 and Supplementary Table 5.

### Data collection and image processing of pores
A dataset of 21,728 movies of pore structures was collected on a 300 kV Titan Krios electron microscope with a K3 direct electron detector and Quantum energy filter (Gatan) using EPU v4 (Thermo Fischer Scientific) and processed in cryoSPARC v3.2.0 (Structura Biotechnology)[49], unless otherwise stated (Supplementary Fig. 9). Electron micrograph movie frames were aligned by MotionCor2[52] in 5×5 patches. CTF parameters were estimated, refined, and corrected using CTFFIND4[53]. A round of manual curation was used to exclude sub-optimal micrographs based on poor CTF resolution (>8 Å). Particles were picked and subjected to several iterations of two-dimensional classification in cryoSPARC v3.2.0[49], selecting only the most homogeneous class averages showing identifiable secondary structure features. 2D classes showed a heterogeneous distribution of oligomer stoichiometries, ranging from the largest 24-fold (7% of end view particles), 23-fold (11%), 22-fold symmetry (52%) to 21-fold (30%). It was evident from raw side-views and 2D classes that Mpf2Ba1 pores also dimerized tail-to-tail in a peculiar conformation. About 1,413,104 selected particles contributed to an ab initio 3D model without imposed symmetry (C1), which showed the barrel of one pore inserted inside the other, and different stoichiometries between the inner and outer pore (Supplementary Fig. 9, 3D classes in red and blue). Despite the most common pore stoichiometry in the 2D classes being a 22-mer, in the 3D reconstructions, the inner ring of each dimer showed a clear and consistent 21-fold symmetry, while the outer ring was a mixture of 22-fold and higher symmetries. To best refine the inner 21-fold symmetric pore while excluding density from the poorly resolved outer ring in the dimer, a mask enclosing only the 21-mer pore of the dimer was created from the full 3D map, using the segmentation tool SEGGER[54] in UCSF Chimera v1.15[55]. Local refinement, coupled with per-particle CTF refinement and with C21 symmetry imposed, was run to iteratively refine the masked sub-volume of the Mpf2Ba1 21-mer. The local resolution of the final 3D reconstruction was calculated using 'blocres'[52] in cryoSPARC[49] with an FSC threshold of 0.5. The resolution of the map was determined using masking-effect corrected Fourier Shell Correlation (FSC) in cryoSPARC v3.2.0[49]. The 3D refined final map was sharpened using deepEMhancer v0.14 "highRes" option from the unfiltered unmasked halfmaps[50]. The pore image processing workflow and parameters are summarized in Supplementary Fig. 9 and Supplementary Table 5.

### Model building, analysis, and validation
Pre-pore model: the X-ray structure of Mpf2Ba1-1167 was fitted into the cryo-EM map after initial rigid body assignment using RIBFIND[56] and successive flexible fitting using Flex-EM[57], both from CCP-EM suite[58]. Model refinement was performed in Coot v0.8.9.2[40] and ISOLDE[59].

Pore model: the pre-pore model was a rigid body fitted into the cryo-EM pore map using the same programs as above. Transmembrane β-hairpins were manually built into the β-barrel densities using Coot v0.8.9.2[40] and iteratively refined using ISOLDE[59] and, to ensure proper geometry, real space refinement in PHENIX v1.19.2[60,61].

For the N-terminal cleavage site protease identification, Procleave_sequence webserver[62] was used providing the sequence of the cleavage site.

The electrostatic potential of a Mpf2Ba1 pore subunit was calculated using PDB2PQR software[63], to prepare the model PDB file for continuum solvation in a .pqr file that was used as input for the DelPhi suite[64,65] to solve the Poisson-Boltzmann equation for the pore subunit.

For both pre-pore and pore models, Q-score from MapQ[66], TEMPy SMOC[57,67], MolProbity[68], and 3D-Strudel[69] scores were used to validate the final model (Supplementary Table 6).

Model-to-map FSCs were estimated using PHENIX v1.19.2[61]. Structure visualization, fitting of multiple subunits, segmentation, analysis of charges and contacts, images, and movies were generated with UCSF Chimera v1.15[55] and ChimeraX v1.2 to 1.4[44].

## Data availability

Sequences for Mpf2Ba1 and Mpf2Ba1-1167 were deposited in NCBI under accession numbers OP537915 (Mpf2Ba1) and OP575916 (Mpf2Ba1-1167). PDB coordinates, and cryo-EM maps are deposited with the Protein Data Bank under accession numbers 8B6U (Mpf2Ba1-1167 monomer), 8B6V and EMD-15882 [https://doi.org/10.2210/pdb8B6V/pdb] (Mpf2Ba1 pre-pore), and 8B6W and EMD-15883 [https://doi.org/10.2210/pdb8b6w/pdb] (Mpf2Ba1 pore). Source data are provided with this paper.

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

## Acknowledgements

We thank the Protein Core Facility, Crop Transformation Systems, and Controlled Environments Group of Corteva Agriscience for help with protein expression, plant transformation, and greenhouse support, respectively; USDA-ARS, Plant Sciences Institute, Beltsville, MD for providing IPLB-DU182A cells; E. Schepers for Edman degradation sequencing; Z. Hou for help in structural analysis; D. Cerf for bioassay of *N. viridula* and *D. speciosa*; C. Stewart and T. P. Stewart, X-ray Facility manager and "Roy J. Carver" High Resolution Microscopy Facility manager at Iowa State University; D. Houldershaw, S. Malhotra, T. Cragnolini, A. Sweeney, L. Genz, and S. Nair for modeling and model prediction support, C. Bagneris for support in the Rosalind Franklin Lab; S. Chen for imaging support, C. Moores and all members of Topf, Saibil and Orlova groups for fruitful scientific discussion at Birkbeck College, University of London. This work was funded by Corteva Agriscience. Cryo-EM data was collected at the ISMB EM facility at Birkbeck College, University of London, with financial support from Corteva Agriscience (grant number 103973-10, granted to H.R.S.) and The Wellcome Trust (202679/Z/16/Z and 206166/Z/17/Z, granted to H.R.S.). Computational analysis was also supported by Leibniz Science Campus InterAct, funded by the BWFGB Hamburg and the Leibniz Association (grant W6/2018, granted to M.T.).

## Author contributions

M.E.N., N.E., M.T., and H.R.S. conceived the idea and provided supervision. G.M., B. P., C.L., A.L., T.M., J.K., J-Z.Z., A.K., E.S., and C.P.O. performed the experiments, C.L., D.S., J.K.B., D.A., N.Y., M.E.N. and N.E. provided the samples, B.P. performed the X-ray crystallography, G.M. and N.L. performed cryo-EM, G.M. performed the single-particle analysis and molecular modeling, A.K. conducted the diet bioassays, J-Z.Z. and A.S. planned and conducted the experiments with field-derived resistant WCR colonies, V.C. supervised vector construction, G.S. and T.M. conducted the greenhouse experiments, T.M., J.K. and T.N. planned and conducted the field experiments, A.L.L. provided supervision and guidance, G.M., B.P., M.E.N., N.E., M.T., and H.R.S. wrote the manuscript.

## Competing interests

We declare that authors B.P., C.L., D.S., J.K.B., D.A., A.L., E.S., C.P.O., N.Y., G.S., T.M., J.K., T.N., J-Z.Z., A.S., A.K., V.C., A.L.L., M.E.N., and N.E.

are or were employed by Corteva Agriscience which provided the funding for this work. The remaining authors declare no competing interest.
