## [Peer Review File · Nature Communications]

Reviewers' Comments:

Reviewer #1:

Remarks to the Author:

The manuscript by Marini and colleagues provides a structural characterization of an insecticidal MACPF domain protein. The protein is composed of two domains, MACPF and beta prism, and is able to bind to lipid membranes, oligomerize and form pores. Characterizing novel insecticidal proteins is needed due to emerging insect resistance. This manuscript provides structural information and molecular details into the pore formation process by the novel insecticidal protein. The manuscript is very well written, the figures are clear and rich with detail. The presented structures are of high quality and will be interesting for a wide audience. I think paper deserved to be published pending minor revisions. I really enjoyed reading this paper. The following minor comments could further improve the manuscript.

The model on Fig. 4 assumes that oligomerization takes place on the membrane surface. Is it possible that monomers oligomerize in the lumen of WCR larval midgut and then bind to the membrane and form pores? Why this possibility is not considered, especially since prepores were obtained in that way upon proteolytic activation.

Can something be said about membrane binding step? Does the shape and properties of the beta prism domain allow conclusions with regards to the membrane binding or even specificity to particular membrane lipids (or proteins)?

Data on Fig. 1c show that protein is Ca²⁺ dependent. Why? Is it connected with binding to membranes? Please explain.

To my understanding the experiment on Fig. 1d is not explained in the materials and methods section. Please include appropriate method section.

Could labelling of protein effected the assay shown in Fig. 1b? What precautions were taken? Please comment.

Is there any reason why was this particular lipid mixture used for pore formation and imaging (POPC, 4% cholesterol)?

Line 569: Is the appropriate reference for DelPhi suite 65?

Reference 39 lacks bibliographic details.

Extended data fig. 3 shows the sequence alignment of only MACPFs and not CDCs.

Reviewer #2:

Remarks to the Author:

The manuscript, "Structural journey of an insecticidal protein against western corn rootworm" by Marini et al. uses X-ray crystallography and cryo-electron microscopy (cryo-EM) to elucidate the three structural states of a novel insecticidal pore-forming protein (Mpf2Ba1) from the membrane attack complex perforin family (MACPF). The general significance of the initial discovery and elucidation of the three-dimensional structures of the Mpf2Ba1 insecticidal protein complex is that the typical *Bacillus thuringiensis* (BT) insecticidal proteins used in modern transgenic corn strains are showing instances of pest resistance in the field; therefore, new insecticidal proteins from non-BT microorganisms are needed with novel modes and sites of action.

The insecticidal protein studied, Mpf2Ba1, was isolated from *Pseudomonas monteilii* in an artificial diet screening process against western corn rootworm (WCR). Interestingly, the purified Mpf2Ba1 protein from *P. monteilii* specifically showed activity against WCR and northern corn rootworm but not other insect pests. The authors used fluorescence assays to reveal that the Mpf2Ba1 protein binds to the WCR midgut and its presumable pore forming activity caused rapid ion permeability. Moreover, when

Mpf2Ba1 was introduced into maize transgenically it provided root protection from rootworm pests. This protection is afforded against WCR isolated from fields of BT maize currently showing a lack of resistance to BT toxins. Hence, the mechanism of action for Mpf2Ba1 is unique from the BT-derived insecticidal toxins.

Once Mpf2Ba1 was established as viable in targeting the pest, WCR, the manuscript then explores the three-dimensional structures of the soluble monomer, the multimerized prepore oligomer, and the beta-barrel pore form. Using X-ray crystallography the authors solve a 2.1 Å structure of the soluble Mpf2Ba1 monomer. The structure reveals Mpf2Ba1 is a two-domain protein with amino-terminal MACPF pore-forming domain and a carboxy-terminal beta-prism domain. The MACPF domain contains two amphipathic beta hairpins, which presumably constitute the pore. The beta prism domain has homology to the insecticidal delta-endotoxins from BT and other proteins. It also contains a coordinated Mg ion the authors speculate is stabilizing.

They then show midgut protein factor is required to assemble the Mpf2Ba1 monomer into an oligomeric prepore. The higher order oligomer was isolated by size-exclusion chromatography and then its structure was imaged by cryo-EM to 3.1 Å. The ring-shaped structure revealed a 22-mer that is 80 Å tall, 260 Å in diameter, with a 120 Å-wide lumen. There were some small, but notable, changes in the prepore oligomer relative to the unassembled, soluble monomer. One major change is that proteolytic activation exposes contacts in the amino-terminal domain that make key inter-domain contacts. Also the carboxy-terminal beta-prism domain rotates 15 degrees relative to its position in the soluble monomeric structure.

Finally, to obtain the pore state of the oligomer the investigators use either high temperature (~50 degrees C) or POPC/cholesterol lipid exposure to drive the conversion of the prepore to pore. Then the structure was determined by cryo-EM to 2.6 Å, yielding the structure of the 21-mer pore. The largest conformational change between prepore and pore is the change in the alpha-helical bundles into 100 Å-long membrane spanning beta-hairpins. This structural motif is common to other pore-forming MACPF family members.

This manuscript is a tour de force structure/function exploration of a newly discovered insecticidal MACPF--the highest resolution structure of a MACPF to date. The manuscript writing is very thorough yet concise considering the functional and structural breadth covered.

(1) Can the authors speculate on why the imaged prepore oligomer was a 22-mer but the imaged pore was a 21-mer? Were these the dominant species in either context or were other minor populations of other oligomeric states observed. While pore formation will not likely be greatly affected by the ultimate oligomeric state, it would address a minor confusing point in the current narrative.

(2) Would have liked to have seen a little more description of the screening process and results that identified Mpf2Ba1 in the first place? How were the bacterial strains selected? How many strains were screened, etc.?

(3) Is the oligomerization factor in WCR midgut a membrane-associated protein or a soluble factor? Could the oligomerization factor also be the receptor, where multiple copies of receptor at the membrane interface bring proteolytically activated Mpf2Ba1 monomers together for oligomerization?

Reviewer #3:

Remarks to the Author:

This manuscript gives a clear and detailed description of the structure an Mpf class of insecticidal protein in monomer, prepore and pore forms with the highest quality structures for this type of protein. The features described are clearly illustrated in the text and supported by informative videos. Activity of the protein against larvae resistant to other control proteins is also demonstrated. This will be beneficial not only to the field of insecticidal proteins and pest control but also to those working in the wider field of pore-forming proteins.

My only substantial criticisms are relatively minor and relate to clarity over which form of the protein (wild-type or variant) is actually being described at each stage, and the need to clarify issues around

activation by gut juice or remove speculation (see below).

- Line 66: “no activity against other major agricultural pest species”, as this is a little vague it would be useful to specify here what was tested (either as data not shown or add to tables) and no tests are described in the methods. Were Southern corn rootworm larvae tested? Fig1c uses SCR cells but are larvae susceptible?

- There may be some lack of clarity in the labelling of Fig2. The text says that the structure of Mpf2Ba1-1167 was solved and shown in this figure, but the figure refers to Mpf2Ba1 (implying the non-mutant form). It is not entirely clear, thereafter, when prepore and pore forms of Mpf2Ba1 are described, whether these are still the mutant forms or whether they are wild-type. Methods line 558 refers to Mpf2Ba1 structure being fitted but only the mutant X-ray structure has been listed (similar line 787, 797 and supp table 4). Lines 666-668 also speak of Mpf2Ba1 where, in at least one instance the mutant may be meant. Please ensure that the variants are described clearly throughout – it may mean repeating longer names but it is then clear which variant is described. This may apply to other parts of the work eg is the event in fig1d wt or mutant? Answers to some of these queries may be found in the methods but would be best addressed directly in the main text.

- Lines 102-108: Cleavage after K25 is reported. It would be interesting to know if there is any further proteolytic cleavage – we are not told explicitly that the rest of the protein is visible in the structure without breaks and this would be useful information to include. If it has breaks or missing C-terminal density, further cleavage might occur, which could be clarified by mass spec analysis.

- Lines 120-123: The speculation that cathepsin D or E are involved is interesting but the evidence provided (see above) does not suggest cleavage beyond K25 (the remaining residues 26 to 31 may just be too disordered). Mass spec or N-terminal sequencing of the prepore form should resolve whether further processing has occurred.

I would modify the way that the putative enzymes are described (if description is retained) -cathepsin E is associated with endothelial layers and is not secreted, so it is not entirely clear how it would access the prepore protein in the lumen. Cathepsin D is lysosomal. The literature outside the field vertebrate enzymes may confuse these names but to get away from this I suggest referring to the activity as cathepsin D-like. In addition, both these enzymes, as aspartic proteinases, can be easily inhibited by pepstatin and this might be a worthwhile experiment so that a role for this class of enzyme could be confirmed or the speculation eliminated. In line 122 reference is made to “cathepsins”. I would also suggest that this is modified as it is unspecific -while the above cathepsins are aspartic proteinases, others eg cathepsin H, are cysteine proteinases. Finally, if aspartic proteinases are present and active in the gut as suggested in reference 23 and consistent with the gut pH of 5.75 reported in that paper, I suspect that trypsin would not also be active in that environment.

- lines 132-133: font change

- The final sentence of Fig4 legend is not necessary and does not really describe the figure. The point is made elsewhere in the text

- Methods describe heating to $\sim 50^{\circ}\text{C}$ (line 492) but the text says $\sim 53^{\circ}\text{C}$ (line 160). Please clarify and use a consistent value.

RESPONSES

Reviewer #1 (Remarks to the Author):

The manuscript by Marini and colleagues provides a structural characterization of an insecticidal MACPF domain protein. The protein is composed of two domains, MACPF and beta prism, and is able to bind to lipid membranes, oligomerize and form pores. Characterizing novel insecticidal proteins is needed due to emerging insect resistance. This manuscript provides structural information and molecular details into the pore formation process by the novel insecticidal protein. The manuscript is very well written, the figures are clear and rich with detail. The presented structures are of high quality and will be interesting for a wide audience. I think paper deserved to be published pending minor revisions. I really enjoyed reading this paper. The following minor comments could further improve the manuscript.

The model on Fig. 4 assumes that oligomerization takes place on the membrane surface. Is it possible that monomers oligomerize in the lumen of WCR larval midgut and then bind to the membrane and form pores? Why this possibility is not considered, especially since pre-pores were obtained in that way upon proteolytic activation.

Response: We agree that we cannot rule out the possibility of oligomerization happening in the lumen, followed by membrane binding and subsequent pore formation. Indeed, we were able to obtain oligomers in solution, but we do not think this is a physiological pathway. In addition, the dimerization of both pre-pore and pore may be a consequence of the artificial environment. Fig. 4 presents our proposed model of Mpf2b1 mechanism of action *in vivo* inside the larval midgut. We have now clarified this in the description of Fig. 4.

Can something be said about membrane binding step? Does the shape and properties of the beta prism domain allow conclusions with regards to the membrane binding or even specificity to particular membrane lipids (or proteins)?

Response: We appreciate the question raised by the reviewer and can only say that we continue to investigate the nature and identity of the membrane receptor but prefer not to speculate here.

Data on Fig. 1c show that protein is Ca²⁺ dependent. Why? Is it connected with binding to membranes? Please explain.

Response: Fig. 1c does not show that the protein is Ca²⁺-dependent, rather it shows that the signal generated depends on extracellular Ca²⁺ as would be expected for a pore forming protein. We added more text to the figure legend for clarity.

To my understanding the experiment on Fig. 1d is not explained in the materials and methods section. Please include appropriate method section.

Response: These pictures were taken during the field-testing experiment that is summarized in Supplementary Fig. 1b. We have added text to make this clear.

Could labelling of protein effected the assay shown in Fig. 1b? What precautions were taken? Please comment.

Response: Fig. 1b shows how the unlabelled protein can displace the labelled protein, so there is no concern regarding the label affecting the assay results. The label is simply a way of identifying the binding site for the protein. If the label alters binding, the unlabelled would not displace it. Again, the experiment measures the occupation of the receptor by the unlabelled protein.

Is there any reason why was this particular lipid mixture used for pore formation and imaging (POPC, 4% cholesterol)?

Response: This mixture has been used for planar lipid bilayer testing of other insecticidal pore forming proteins (e.g., Perez Ortega C, Leininger C, Barry J, Poland B, Yalpani N, Altier D, Nelson ME, Lu AL. 2021. Coordinated binding of a two-component insecticidal protein from *Alcaligenes faecalis* to western corn rootworm midgut tissue. *J Invertebr Pathol* 183:107597). We therefore thought that it would be appropriate for testing Mpf2Ba1.

Line 569 (now 533): Is the appropriate reference for DelPhi suite 65?

Response: We thank the reviewer for noticing that! We added the appropriate references for DelPhi web server (65. Smith et al., 2012; 66. Sarkar et al., 2013).

Reference 39 lacks bibliographic details.

Response: The reviewer is correct. Reference 39 was under revision during the review of this manuscript, and we assumed that it would be accepted in time to add the details. However, that has not occurred, so we have removed it from the manuscript and refer to the data provided in the supplemental information instead (Supplementary Table 3).

Extended data fig. 3 shows the sequence alignment of only MACPFs and not CDCs.

Response: This was our mistake, we thank the reviewer for pointing it out! We have now changed it to MACPFs only.

Reviewer #2 (Remarks to the Author):

The manuscript, "Structural journey of an insecticidal protein against western corn rootworm" by Marini et al. uses X-ray crystallography and cryo-electron microscopy (cryo-EM) to elucidate the three structural states of a novel insecticidal pore-forming protein (Mpf2Ba1) from the membrane attack complex perforin family (MACPF). The general significance of the initial discovery and elucidation of the three-dimensional structures of the Mpf2Ba1 insecticidal protein complex is that the typical *Bacillus thuringiensis* (BT) insecticidal proteins used in modern transgenic corn strains are showing instances of pest resistance in the field; therefore, new insecticidal proteins from non-BT microorganisms are needed with novel modes and sites of action.

The insecticidal protein studied, Mpf2Ba1, was isolated from *Pseudomonas montellii* in an artificial diet screening process against western corn rootworm (WCR). Interestingly, the purified Mpf2Ba1 protein from *P. montellii* specifically showed activity against WCR and northern corn rootworm but not other insect pests. The authors used fluorescence assays to reveal that the Mpf2Ba1 protein binds to the WCR midgut and its presumable pore forming activity caused rapid ion permeability. Moreover, when Mpf2Ba1 was introduced into maize transgenically it provided root protection from rootworm pests. This protection is afforded against WCR isolated from fields of BT maize currently showing a lack of resistance to BT toxins. Hence, the mechanism of action for Mpf2Ba1 is unique from the BT-derived insecticidal toxins.

Once Mpf2Ba1 was established as viable in targeting the pest, WCR, the manuscript then explores the three-dimensional structures of the soluble monomer, the multimerized prepore oligomer, and the beta-barrel pore form. Using X-ray crystallography the authors solve a 2.1 Å structure of the soluble Mpf2Ba1 monomer. The structure reveals Mpf2Ba1 is a two-domain protein with amino-terminal MACPF pore-forming domain and a carboxy-terminal beta-prism domain. The MACPF domain contains two amphipathic beta hairpins, which presumably constitute the pore. The beta prism domain has homology to the insecticidal delta-endotoxins from BT and other proteins. It also contains a coordinated Mg ion the authors speculate is stabilizing.

They then show midgut protein factor is required to assemble the Mpf2Ba1 monomer into an oligomeric prepore. The higher order oligomer was isolated by size-exclusion chromatography and then its structure was imaged by cryo-EM to 3.1 Å. The ring-shaped structure revealed a 22-mer that is 80 Å tall, 260 Å in diameter, with a 120 Å-wide lumen. There were some small, but notable, changes in the prepore oligomer relative to the unassembled, soluble monomer. One major change is that proteolytic activation exposes contacts in the amino-terminal domain that make key inter-domain contacts. Also the carboxy-terminal beta-prism domain rotates 15 degrees relative to its position in the soluble monomeric structure.

Finally, to obtain the pore state of the oligomer the investigators use either high temperature (~50 degrees C) or POPC/cholesterol lipid exposure to drive the conversion of the prepore to pore. Then the structure was determined by cryo-EM to 2.6 Å, yielding the structure of the 21-mer pore. The largest conformational change between prepore and pore is the change in the alpha-helical bundles into 100 Å-long membrane spanning beta-hairpins. This structural motif is common to other pore-forming MACPF family members.

This manuscript is a tour de force structure/function exploration of a newly discovered insecticidal MACPF--the highest resolution structure of a MACPF to date. The manuscript writing is very thorough yet concise considering the functional and structural breadth covered.

(1) Can the authors speculate on why the imaged prepore oligomer was a 22-mer but the imaged pore was a 21-mer? Were these the dominant species in either context or were other minor populations of other oligomeric states observed. While pore formation will not likely be greatly affected by the ultimate oligomeric state, it would address a minor confusing point in the current narrative.

Response: We thank the reviewer for giving us the opportunity to clarify this point. The pre-pores stoichiometries showed less variability and a clear predominance of 22-mers, while the pore stoichiometries ranged from 21-mers to 24-mers, clearly visible and manually counted in end views

2D classes (Supplementary Fig. 8c). We now explain our choice to focus on the 3D structure of the 21-fold symmetric pore, instead of 22-fold as for the pre-pore, in the main text (lines 186-193) and in more detail in the Methods (lines 508-514).

(2) Would have liked to have seen a little more description of the screening process and results that identified Mpf2Ba1 in the first place? How were the bacterial strains selected? How many strains were screened, etc.?

Response: The screening process was identical to what has been thoroughly described in *Schellenberger et al. 2016* and *Wei et al., 2018*, which we referenced in Supplementary Methods (line 764, reff. 71,72). We feel that it is sufficient to refer readers to those papers, if interested.

(3) Is the oligomerization factor in WCR midgut a membrane-associated protein or a soluble factor? Could the oligomerization factor also be the receptor, where multiple copies of receptor at the membrane interface bring proteolytically activated Mpf2Ba1 monomers together for oligomerization?

Response: The factor is soluble in gut fluid extracts. We agree with the reviewer's thinking that the factor could be the membrane receptor (or part of it) that has been released into gut fluid. We described what we have done to characterize its physical properties in the MS (lines 113-115). We continue to try to identify the factor.

Reviewer #3 (Remarks to the Author):

This manuscript gives a clear and detailed description of the structure an Mpf class of insecticidal protein in monomer, prepore and pore forms with the highest quality structures for this type of protein. The features described are clearly illustrated in the text and supported by informative videos. Activity of the protein against larvae resistant to other control proteins is also demonstrated. This will be beneficial not only to the field of insecticidal proteins and pest control but also to those working in the wider field of pore-forming proteins. My only substantial criticisms are relatively minor and relate to clarity over which form of the protein (wild-type or variant) is actually being described at each stage, and the need to clarify issues around activation by gut juice or remove speculation (see below).

- Line 66 (now 67): “no activity against other major agricultural pest species”, as this is a little vague it would be useful to specify here what was tested (either as data not shown or add to tables) and no tests are described in the methods. Were Southern corn rootworm larvae tested? Fig1c uses SCR cells but are larvae susceptible?

Response: Corn earworm, fall armyworm, European corn borer, soybean looper, and southern green stinkbug were tested at ≥ 400 ppm and no effects were observed. We have added this to the text at lines 67-70. SCR (*D. undecimpunctata*) was also tested but did not show any effects at 1250 ppm. We continue to try to understand why SCR are not affected, while the Du182A cells are sensitive, but have clarified this finding in the Methods section (lines 374-376).

- There may be some lack of clarity in the labelling of Fig2. The text says that the structure of Mpf2Ba1-1167 was solved and shown in this figure, but the figure refers to Mpf2Ba1 (implying the non-mutant form). It is not entirely clear, thereafter, when prepore and pore forms of Mpf2Ba1 are described, whether these are still the mutant forms or whether they are wild-type. Methods line 558 (now 522) refers to Mpf2Ba1 structure being fitted but only the mutant X-ray structure has been listed (similar line 787, 797 (now 1037 and 1047) and supp table 4). Lines 666-668 (now 543-544) also speak of Mpf2Ba1 where, in at least one instance the mutant may be meant. Please ensure that the variants are described clearly throughout – it may mean repeating longer names but it is then clear which variant is described. This may apply to other parts of the work eg is the event in fig1 wt or mutant? Answers to some of these queries may be found in the methods but would be best addressed directly in the main text.

Response: We agree and thank the reviewer for pointing this out. A label referring to the Mpf2Ba1-1167 variant has been added to Fig. 2 and whenever appropriate throughout the manuscript and Supplementary data.

- Lines 102-108 (now 112-118): Cleavage after K25 is reported. It would be interesting to know if there is any further proteolytic cleavage – we are not told explicitly that the rest of the protein is visible in the structure without breaks and this would be useful information to include. If it has breaks or missing C-terminal density, further cleavage might occur, which could be clarified by mass spec analysis.

Response: We thank the reviewer for allowing us to specify that Mpf2Ba1 model fits in the prepore map until the very last C-term residue without breaks. We therefore exclude further proteolytic cleavage beyond Glu30 and explicitly state this in the manuscript (line 127).

- Lines 120-123 (now 133-137): The speculation that cathepsin D or E are involved is interesting but the evidence provided (see above) does not suggest cleavage beyond K25 (the remaining residues 26 to 31 may just be too disordered). Mass spec or N-terminal sequencing of the prepore form should resolve whether further processing has occurred.

Response: We do not think that mass spec is needed here since we confirmed that the C-term is intact, and no other breaks are observed in the pre-pore or pore. We repeated the N-term sequencing of the oligomer (purified) and the gut fluid processed monomer: the cleavage site is still after K25.

I would modify the way that the putative enzymes are described (if description is retained) - cathepsin E is associated with endothelial layers and is not secreted, so it is not entirely clear how it would access the prepore protein in the lumen. Cathepsin D is lysosomal. The literature outside the field vertebrate enzymes may confuse these names but to get away from this I suggest referring to the activity as cathepsin D-like.

Response: We agree, and we now refer to the putative protease as cathepsin D-like (lines 133 and 136).

In addition, both these enzymes, as aspartic proteinases, can be easily inhibited by pepstatin and this might be a worthwhile experiment so that a role for this class of enzyme could be confirmed or the speculation eliminated.

Response: We found that treating WCR gut fluid with a protease inhibitor cocktail or PMSF alone could prevent N-terminal processing leading to oligomerization consistent with processing by serine proteases. This information has been added to the Supplementary Results & Discussion (lines 952-955). Pepstatin was not tested because of the result for PMSF. Regardless, we cannot be certain that processing *in vivo* does not occur at Thr31 which is the reason for raising the possibility when those residues were absent from the pre-pore map. Moreover, the absence of 6 amino acids in question in the map likely indicates that their presence or absence does not affect pre-pore assembly. We hope that the reviewer finds our interest in keeping the information in the manuscript acceptable.

In line 122 (now 136) reference is made to "cathepsins". I would also suggest that this is modified as it is unspecific -while the above cathepsins are aspartic proteinases, others eg cathepsin H, are cysteine proteinases.

Response: We agree again, and we now specify that we refer to aspartic proteases belonging to the cathepsins family (lines 136).

Finally, if aspartic proteinases are present and active in the gut as suggested in reference 23 and consistent with the gut pH of 5.75 reported in that paper, I suspect that trypsin would not also be active in that environment.

Response: As we mentioned above in response to pepstatin/proteases inhibition, a possible explanation for the discrepancy could be that processing *in vivo* is different than our *in vitro* result, partially due to pH differences as the Reviewer questions. Other factors include ionic or osmotic effects or membrane interaction. However, the presence or absence of the 6 additional amino acids doesn't appear to play a role in pre-pore assembly. We hope that the Reviewer agrees with this conclusion and supports our interest in keeping the information in the manuscript.

• lines 132-133 (now 149-150): font change

Response: Thank you, it is now correctly formatted.

• The final sentence of Fig4 legend is not necessary and does not really describe the figure. The point is made elsewhere in the text.

Response: We agree and removed it.

• Methods describe heating to ~50°C (line 492 – now 457) but the text says ~53°C (line 160 – now 180). Please clarify and use a consistent value.

Response: Thank you, we corrected those values to the temperature range tested (~50-53°C).

REVIEWERS' COMMENTS

Reviewer #3 (Remarks to the Author):

The authors have provided clear and appropriate responses to my previous comments and, as far as I can see, to the comments of other reviewers. I look forward to seeing this work in print and have no further issues to address